# Acute Heat Stress Induces the Differential Expression of Heat Shock Proteins in Different Sections of the Small Intestine of Chickens Based on Exposure Duration

**DOI:** 10.3390/ani10071234

**Published:** 2020-07-21

**Authors:** Sharif Hasan Siddiqui, Darae Kang, Jinryong Park, Hyun Woo Choi, Kwanseob Shim

**Affiliations:** 1Department of Animal Biotechnology, College of Agriculture and Life Sciences, Jeonbuk National University, Jeonju 54896, Korea; hasanshuhin@gmail.com (S.H.S.); kangdr92@gmail.com (D.K.); wlsfyd1321@naver.com (J.P.); 2Department of Animal Science, College of Agriculture and Life Sciences, Jeonbuk National University, Jeonju 54896, Korea

**Keywords:** acute heat stress, chicken, heat shock protein, small intestine

## Abstract

**Simple Summary:**

The small intestine is a pivotal organ in the alimentary tract of chickens. Approximately 90% of digestion and absorption occurs in the small intestine. The expression of heat shock proteins (HSPs) in different sections of the small intestine upon varying durations of heat exposure is unclear. In this study, we evaluated the protein and mRNA expression levels of *HSP70*, *HSP60*, and *HSP47* in the duodenum, jejunum, and ileum, on acute heat exposure. The protein and gene expression of the HSPs in the different sections of the small intestine of chicken increased at different stages of acute heat stress and were reduced after the induction of heat tolerance. However, acute heat stress damaged the integrity of the different sections of the small intestine and liver in chickens.

**Abstract:**

In this study, we examined the protein and gene expression of heat shock proteins (HSPs) in different sections of the small intestine of chickens. In total, 300 one-day-old Ross 308 broiler chicks were randomly allocated to the control and treatment groups. The treatment group was divided into four subgroups, according to the duration of acute heat exposure (3, 6, 12, and 24 h). The influence of heat stress on the protein and gene expression of *HSP70*, *HSP60*, and *HSP47* in different sections of the small intestine of chickens was determined. The protein expression of *HSP70* and *HSP60* was significantly higher at 6 h in the duodenum and jejunum and 12 h in the ileum. The *HSP47* protein expression was significantly higher at 3 h in the duodenum and ileum and at 6 h in the jejunum. The gene expression levels of *HSP70*, *HSP60*, and *HSP47* were significantly higher at the 3 h treatment group than the control group in the duodenum, jejunum, and ileum. The glutamate pyruvate transaminase and glutamate oxaloacetate transaminase levels were significantly higher at 12 and 24 h in the serum of the blood. Acute heat stress affected the expression of intestinal proteins and genes in chickens, until the induction of heat tolerance.

## 1. Introduction

Climate change affects the poultry industry in tropical countries, and even in regions with a milder climate. The challenges during the summer include decreased growth performance and increased morbidity and mortality, which ultimately result in economic losses [1,2]. Heat stress is one of the most challenging environmental and industrial hazards, especially in chickens. Moreover, heat stress causes physiological and serological changes in chickens [3], which severely damage organs or tissues. Previous studies have indicated that the effects of heat stress on reduced organ function and the health of chickens can be confirmed by evaluating the changes in various serological parameters, such as serum glutamate pyruvate transaminase (SGPT) and serum glutamate oxaloacetate transaminase (SGOT) [4,5,6].

Acute heat stress induces the reduction of feed intake, and digestibility, and reduces the function of different organs in animals, resulting in the suppression of the growth rate [7,8,9]. Heat shock proteins (HSPs) are indicators of acute heat stress. The HSPs are highly conserved proteins that stimulate cytoprotection and induce heat tolerance in the cells of various organs and tissues, during the heat stress period [10,11]. The HSPs also maintain the integrity of organs by repairing damaged or misfolded proteins and promoting cell survival [12]. Heat-stress-induced transcription factors bind to the promoters of heat-shock-related genes, enhances the expression of HSP genes, and consequently the production of HSPs [13]. One of the HSPs, *HSP70*, which is involved in the cellular response to different environmental stressors, might be a biomarker of the stress response [14]. The expression of HSP60 is considered a “danger signal” in stressed or damaged cells, and particularly the pro-inflammatory response in the cellular innate immune system [15]. In addition, the *HSP47* is a collagen-specific chaperone and a universal biomarker in collagen-inducing cells during wound healing [16]. The expression of *HSP47* is correlated with the aging of the organism and the duration of heat-stress exposure [17].

The gastrointestinal (GI) tract of chickens has a vast surface area, which acts as a regulatory barrier for microbes and exogenous pathogens. The small intestine is the longest and most pivotal part of the GI tract and is located between the stomach and the large intestine. The duodenum, jejunum, and ileum of the small intestine are involved in nutrient transport, digestion, and absorption, providing substances for growth and organ development [18]. Approximately 90% of digestion and absorption occurs in the small intestine, and the remainder occurs in the stomach and the large intestine. The small intestine consists of a continuous layer of squamous epithelial cells. This allows the GI tract to respond promptly to heat stress, which changes the normal structure of the intestine [19] by affecting the intestinal epithelial integrity [20]. Several studies showed the expression pattern of HSPs in the intestine on heat or other stresses [21,22]. However, to our knowledge, no study has evaluated the expression of *HSP70*, *HSP60*, and *HSP47* in the small intestine of chickens on acute heat stress exposure.

Here, we examined the protein and mRNA expression levels of *HSP70*, *HSP60*, and *HSP47* in the duodenum, jejunum, and ileum of broiler chicks, upon exposure to acute heat stress for different durations of time. Additionally, damage to the liver tissues was also observed by measuring the SGPT and SGOT levels in blood serum.

## 2. Materials and Methods

### 2.1. Birds and Experimental Design

The care and use of the chickens complied with the institutional guidelines, and the animal study protocol was approved by the Animal Experiment Administration Committee of the Jeonbuk National University, Jeonju, Republic of Korea (approval number: CBNU2018-097). All efforts were made to reduce discomfort, pain, and misery to the birds.

Three hundred 1-day-old Ross 308 broiler chicks were purchased from a local hatchery at Iksan, Republic of Korea. The chicks were maintained in a battery cage at an environmentally controlled farm, with continuous white light throughout the experiment. The farm temperature was maintained at 33 °C for the first 3 days and then reduced to 30 °C on days 4 to 6. Thereafter, the temperature was reduced by 2 °C per week to a final temperature of 26 ± 1 °C by day 20. In total, 128 chickens aged 21 days weighing approximately 1 kg were randomly selected and allocated to a control group (no heat stress, *n* = 8) or the treatment group (*n* = 120). The chickens in the treatment group were divided into four subgroups, based on the duration of heat exposure (3, 6, 12, and 24 h). The treatment groups featured 8 replications (8 cages) with 15 birds per cage, for a total of 120 birds. At each time point, one bird was selected from each cage (a total of eight birds at each time point). At 21 days, the chickens in the treatment groups were maintained at 34 ± 1 °C for 24 h. The relative humidity was maintained at around 50%, during heat stress, in the treatment groups. The chickens maintained in the cage were provided feed and water ad libitum throughout the experimental period.

### 2.2. Sample Collection

At 0, 3, 6, 12, and 24 h of acute heat stress, one bird was randomly selected from each cage (eight chickens per group) and sacrificed by cervical dislocation, immediately after the collection of blood. The duodenum, jejunum, and ileum portions of the small intestine were dissected (the duodenum attached to the surface of the pancreas separates the duodenum from the jejunum; Meckel’s diverticulum separates the ileum and jejunum). The dissected intestinal sections were washed, flushed with phosphate-buffered saline, immediately frozen in liquid nitrogen, and stored at −80 °C, until use in the Western blotting and quantitative real-time polymerase chain reaction (qRT-PCR) analyses. These analyses (Western blotting and qRT-PCR) were performed using eight replications (*n* = 8 chickens per group).

### 2.3. Serum Analyses

Each blood sample was collected from the wing vein int o non-heparinized tubes, immediately before sacrifice. The blood sample was centrifuged at 3000× *g* for 15 min at 4 °C, to obtain the serum. The serum was stored at −80 °C until analysis. The levels of SGPT and SGOT were measured using colorimetric commercial kits (Asan Pharmaceutical, Co. Ltd., Seoul, Republic of Korea), following the manufacturer’s instructions.

### 2.4. Western Blotting

Heat stress induces different HSPs that protect cells by increasing their heat tolerance capacity. Western blotting was performed to analyze the effect of different durations of acute heat exposure on the protein expression levels of *HSP70*, *HSP60*, and *HSP47* in the duodenum, jejunum, and ileum of chickens. One hundred milligrams of the duodenum, jejunum, and ileum tissue were each lysed in Radioimmunoprecipitation assay (RIPA) buffer (150 mM NaCl, 1% Triton X-100, 0.1% sodium dodecyl sulfate (SDS), 1% sodium deoxycholate, 50 mM Tris-HCl (pH 7.5), and 2 mM EDTA) containing protease inhibitor (Invitrogen, Carlsbad, CA, USA), by vortexing for 30–60 s. Each sample was centrifuged at 15,000 × *g* for 15 min at 4 °C. The supernatant was collected for protein estimation. The protein concentration in the supernatant was evaluated using the DC Protein Assay kit (Bio-Rad, Hercules, CA, USA). The samples were heated in a sample buffer (Elpis, Taejeon, Republic of Korea) at 95 °C for 10 min, and then cooled to 4 °C. The samples (30 μg) were subjected to SDS-polyacrylamide gel electrophoresis (PAGE) using 12% gel (Bio-Rad). The resolved proteins were electroblotted onto a polyvinylidene difluoride (PVDF) membrane (Bio-Rad). The membrane was blocked with 5% skim milk prepared in Tris-buffered saline containing Tween 20 (TBST) (TBST; 0.05% Tween 20, 100 mM Tris-HCl, and 150 mM NaCl, pH 7.5), for 1 h at 25 °C. The membrane was incubated with primary antibodies at 4 °C overnight, washed three times with TBST (10 min per wash), and incubated with horseradish peroxidase (HRP)-conjugated secondary antibody, for 1 h, at room temperature. The antibody-specific protein bands were visualized using the chemiluminescent substrate (Invitrogen) in an iBright CL1000 device (Thermo Fisher Scientific, Waltham, MA, USA). The expression levels of *HSP70*, *HSP60*, and *HSP47* were normalized to those of β-actin. The mouse monoclonal anti-β-actin primary antibody (Santa Cruz Biotechnology, Dallas, TX, USA) and anti-*HSP70*, anti-*HSP60*, and anti-*HSP47* antibodies (Enzo Life Science, Farmingdale, NY, USA) were all diluted at 1:2500 for Western blotting. The dilution of the goat anti-mouse IgG (Enzo Life Science) secondary antibody was 1:5000.

### 2.5. RNA Extraction and qRT-PCR

The mRNA levels of *HSP70*, *HSP60*, and *HSP47* in the different sections of the small intestine were analyzed by qRT-PCR [23]. The sequences of the target genes and reference gene (glyceraldehyde 3-phosphate dehydrogenase, *GAPDH*) used for qRT-PCR are presented in Table 1. Total RNA was extracted using the Trizol reagent (Invitrogen), following the manufacturer’s instructions. The concentration and quality (260/280 nm) of the RNA was determined by measuring the absorbance at 230, 260, and 280 nm, using a Nanodrop spectrophotometer (Thermo Fisher Scientific). One microgram of total RNA was reverse transcribed into cDNA, using the iScript™ cDNA synthesis kit (Bio-Rad), following the manufacturer’s instructions. The qRT-PCR analysis of the target and *GAPDH* genes was performed using the SYBR Green^®^ Supermix (Bio-Rad) on a CFX96™ Real-Time PCR Detection System (Bio-Rad). The PCR conditions were as follows—95 °C for 30 s, followed by 40 cycles of 95 °C for 5 s and 58 °C for 5 s, with a melting temperature of 65–95 °C for 5 s. The fold changes in the mRNA levels were determined using the ΔΔCt method. The qRT–PCR results were analyzed using the ΔCt value (Ct gene of interest—Ct *GAPDH* for each sample). The relative gene expression was obtained using the ΔΔCt method (ΔCt sample—ΔCt calibrator), with the control group used as a calibrator to compare treatment sample gene expression, where the relative gene expression was expressed as fold change = 2^−ΔΔCt^ and 2^−ΔΔCt^ [24].

### 2.6. Statistical Analyses

All experimental data were analyzed using SAS version 9.3 (SAS Institute Inc., Cary, NC, USA). The data are represented as the mean ± standard error from eight replications for each experimental group. Duncan’s multiple range test followed by analysis of variance (ANOVA) was used to compare the groups exposed to heat stress for 0, 3, 6, and 24 h. Differences were considered statistically significant when the *p* value was <0.05.

## 3. Results

### 3.1. Effect of Acute Heat Stress on the mRNA Expression Levels of the HSPs

The expression levels of *HSP70*, *HSP60*, and *HSP47* in the duodenum were significantly up-regulated at 6 h on acute heat stress, and were reduced after 6 h (Figure 1A–C). However, the expression levels of *HSP70* and *HSP47* were significantly lower at 24 h than at 3, 6, and 12 h. The expression levels of *HSP60* were significantly lower at 24 h than at 6 and 12 h (*p* < 0.05).

The results of qRT–PCR analysis in the jejunum on acute heat stress showed that the gene expression levels of *HSP70* was significantly up-regulated at 12 h after acute heat stress (*p* < 0.05) (Figure 2A). The gene expression levels of *HSP60* were significantly higher at 6 h than at 0, 3, 12, and 24 h (*p* < 0.05) (Figure 2B). Interestingly, the gene expression level of *HSP47* was significantly up-regulated at 6 h and high levels of *HSP47* expression were maintained until 12 h (Figure 2C). The expression levels of all three genes (*HSP70*, *HSP60*, and *HSP47*) in the jejunum on acute heat stress (3, 6, 12, 24 h) were significantly higher than that in the control group (*p* < 0.05).

The gene expression levels of *HSP70* in the ileum were significantly up-regulated at 3 h with acute heat stress and the up-regulated HSP levels were maintained until 12 h, with acute heat stress (*p* < 0.05) (Figure 3A). The gene expression levels of *HSP60* gradually increased after 3 h and were at their highest at 6 h (Figure 3B). The gene expression levels of *HSP47* were significantly higher at 6 h than at 0, 12, and 24 h (Figure 3C).

### 3.2. Effect of Acute Heat Stress on the Protein Expression Levels of the HSPs

We conducted Western blotting for analysis of HSP protein levels in small intestine on acute heat stress. The protein expression levels of *HSP70* and *HSP60* in the duodenum were significantly up-regulated (*p* < 0.05) at 6 h (Figure 4A,B), and reduced after 6 h. However, the protein expression levels of *HSP47* were significantly up-regulated at 3 h (Figure 4C). Remarkably, the expression levels of *HSP47* were reduced after 6 h, and low protein expression levels were maintained until 24 h (Figure 4C). Figure 4D showed the results of Western blotting band analysis (Figure 4D). Overall, the expression levels of the three HSPs in the acute heat stress groups were significantly higher than that in the control group.

The protein expression levels of *HSP70*, *HSP60*, and *HSP47* in the jejunum were significantly higher after 6 h of acute heat stress treatment than at the other time-points (*p* < 0.05) (Figure 5A–C). Figure 5D showed the results of Western blotting band for *HSP70*, *HSP60*, and *HSP47* in the jejunum. The expression levels of *HSP70* were significantly up-regulated (*p* < 0.05) at 6, 12, and 24 h (Figure 5A). The expression levels of *HSP60* and *HSP47* in the groups of acute heat stress were significantly higher than that in the control group.

The expression levels of *HSP70*, *HSP60*, and *HSP47* significantly increased at the early time points (3 h, 6 h) of acute heat stress (*p* < 0.05) (Figure 6A–D). The protein expression levels of *HSP70* and *HSP60* were significantly higher (*p* < 0.05) at 12 h than at 0, 3, 6, and 24 h (Figure 6A,B). The expression level of *HSP47* was significantly higher at 3 h than at 0, 6, 12, and 24 h (Figure 6C). Figure 6D showed the results of the Western blotting data.

### 3.3. Effect of Acute Heat Stress on the Levels of SGPT and SGOT

The SGPT and SGOT levels are shown in Figure 7A,B. The levels of SGPT were significantly up-regulated after 12 h of acute heat stress (*p* < 0.05). The expression levels of SGOT were significantly higher (*p* < 0.05) at 24 h than at 0, 3, 6, and 12 h (Figure 7B).

## 4. Discussion

Acute heat stress is a non-specific stress that affects livestock, especially chickens, resulting in poor productivity and death. In response to acute heat stress, several metabolic [25] and enzymatic changes were found in the plasma of chicken [26]. The SGPT and SGOT enzymes are predominantly expressed in the liver and at low levels in muscle cells. These enzymes enter the bloodstream and exist at a high level in the damaged liver. The SGPT enzyme catalyzes the transformation of proteins to produce energy in liver cells. The function of the SGOT enzyme is involved in the metabolism of amino acids. Previous study reported that human bloods contained elevated levels of SGPT and SGOT enzymes, upon exposure to temperatures of 41.6 °C and 42 °C, respectively [27]. The levels of SGPT and SGOT in dogs were also increased upon exposure to high temperatures [28]. A study on heat stress reported that heat stress damaged hepatic cells in liver, increased the levels of SGPT and SGOT in blood [29], and changed liver parenchymal cells [30]. In the present study, the blood in chickens exposed to acute heat stress contained high levels of SGPT and SGOT than that in the control group (Figure 7). The acute injury of liver affects intestinal homeostasis. It was reported that the injury of liver affected the integrity of the intestine [31]. Moreover, these results indicated that the hepatic cells were damaged and parenchymal cells were modified upon exposure to acute heat stress.

The HSPs are a family of proteins, expressed in all organisms under stress conditions. The expression levels of HSPs indicate the heat stress level and stress tolerance capacity. The HSP proteins are classified into four families—small HSPs, *HSP60*, *HSP70*, and *HSP90* [32]. A study on the *HSP70* protein was reported and its function and properties were reported. The *HSP70* promotes the correct folding of nascent and misfolded proteins. In stress conditions, *HSP70* protects the damage of the cell [33,34] by enhancing protein expression [35,36,37]. The production of *HSP70* is promoted by heat stress in various tissues, such as the brain, liver, lung, heart, and leukocytes [38,39]. Moreover, the *HSP70* protects the GI epithelium, promotes the absorption and metabolism of nutrients [32,33,34], and exhibits an antioxidant effect [40,41]. Additionally, the *HSP70* accelerates the healing in damaged tissues or cells, by inducing cell proliferation and protein synthesis [42]. Our results demonstrated that the protein expression levels of *HSP70* were significantly increased in the duodenum, jejunum, and ileum of chickens, under acute heat stress. The groups of acute heat stress exhibited higher levels of *HSP70* mRNA in the small intestine (except in the duodenum) than the control group at 12 and 24 h. In contrast, the protein expression levels of *HSP70* in groups of acute heat stress were reduced at the end of the experiment, compared to that in the control group. Previous studies also demonstrated that the protein expression of *HSP70* was increased in the ileum and jejunum of rats and chickens, and there was no change of expression in the duodenum [43,44].

The *HSP60* exists in all types of organisms and promotes the folding, translocation, and assembly of native proteins. The *HSP60* functions as a molecular chaperone and is involved in the activity of other chaperones [45]. Notably, *HSP60* binds to unfolded protein molecules, promotes protein folding, and consequently prevents protein aggregation [46]. Therefore, the gene expression of *HSP60* in different tissues is dependent on stress conditions [47]. The study of *HSP60* showed that the gene expression of *HSP60* was enhanced under heat stress conditions and decreased after reaching a peak [48]. This study demonstrated that the protein and mRNA expression levels of *HSP60* in the ileum, jejunum, and the duodenum of the treatment groups were significantly higher than that of the control group. The jejunal and duodenal protein levels of *HSP60* in the treatment groups decreased gradually, compared to that in the control group at the end of the experiment.

The protein and mRNA levels of *HSP47* in the duodenum, jejunum, and ileum were upregulated after the chickens were exposed to acute heat stress for different durations of time. The protein and gene expression of *HSP47* varied in the different sections of the chicken small intestine. The protein and mRNA expression levels of *HSP47* were markedly increased upon short-term exposure on heat stress and reduced upon prolonged exposure on acute heat stress. Moreover, the treatment groups exhibited a marked decline in the protein expression of *HSP47* in the jejunum, compared to the control group at the end of the experiment. The *HSP47* contributed to the quality control of collagens in the small intestine of chicken in heat stress conditions. Generally, heat stress increases intestinal permeability by disrupting the intestinal integrity [49]. Tang and colleagues reported that the gene expression of *HSP47* in the kidneys of chickens was increased upon short-term exposure to heat stress, and reduced when the chickens were exposed to heat stress for more than 5 h [50]. This result indicated that the expression of HSPs was decreased by enhanced heat tolerance capacity after a certain period of heat stress. Generally, mRNA is translated into protein. However, protein expression and gene expression are unrelated, because the levels of proteins can be changed through mRNA turnover [51]. Therefore, proteins and genes did not show the same expression level in this study, although their expression trend was almost the same. This trend of expression indicates that short-term heat stress induces the production of HSPs that contribute to the restoration of normal cell function, which is reflected by the reduction of the gene expression of HSPs in the later stages of acute heat stress [52].

## 5. Conclusions

Our findings suggest that acute heat stress damages the integrity of the different sections of the small intestine in chickens. The gene and protein expression of *HSP70*, *HSP60*, and *HSP47* in the different sections of the small intestine of chickens, changes according to the duration of acute heat stress. After a certain period of heat exposure, chickens get heat tolerance capacity, which might decrease the HSPs expression levels. Further studies are needed to elucidate the exact mechanism involved.

## Figures and Tables

**Figure 1 animals-10-01234-f001:**
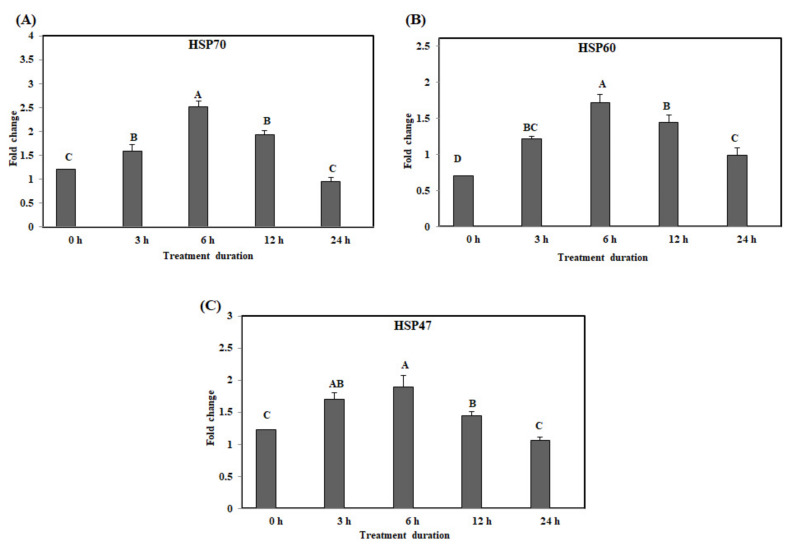
The mRNA expression of the heat shock proteins in the duodenum of chickens exposed to acute heat stress for different durations of time (*n* = 8). The mean relative expression of *HSP70* (**A**), *HSP60* (**B**), and *HSP47* (**C**). ^A–D^
*HSP70*, *HSP60*, and *HSP47* in the columns with different superscript letters differ significantly (*p* < 0.05) among the different heat exposure durations.

**Figure 2 animals-10-01234-f002:**
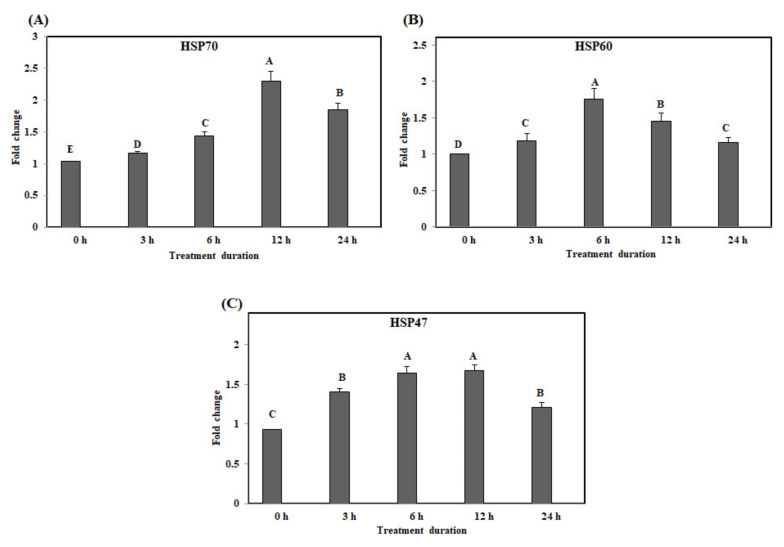
The mRNA expression of the heat shock proteins in the jejunum of chickens exposed to acute heat stress for different durations of time (*n* = 8). The mean relative expression of *HSP70* (**A**), *HSP60* (**B**), and *HSP47* (**C**). ^A–E^
*HSP70*, *HSP60*, and *HSP47* in the columns with different superscript letters differ significantly (*p* < 0.05) among the different heat exposure durations.

**Figure 3 animals-10-01234-f003:**
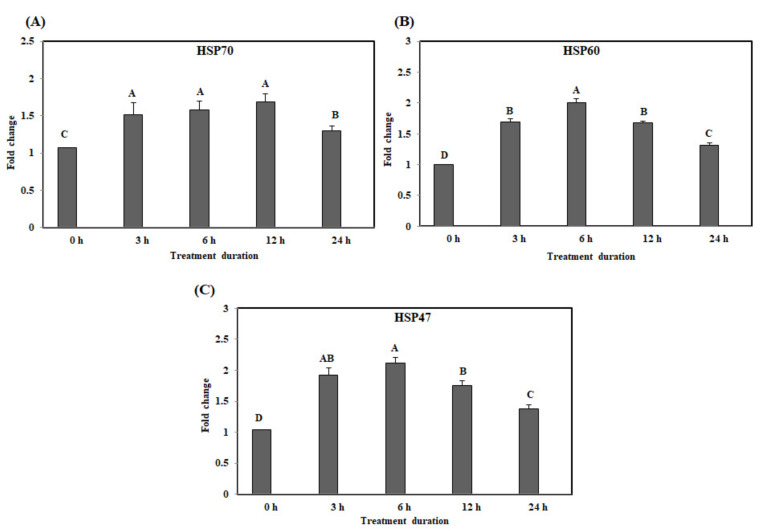
The mRNA expression of the heat shock proteins in the ileum of chickens exposed to acute heat stress for different durations of time (*n* = 8). The mean relative expression of *HSP70* (**A**), *HSP60* (**B**), and HSP47 (**C**). ^A–D^
*HSP70*,*HSP60*, and *HSP47* in the columns with different superscript letters differ significantly (*p* < 0.05) among the different heat exposure durations.

**Figure 4 animals-10-01234-f004:**
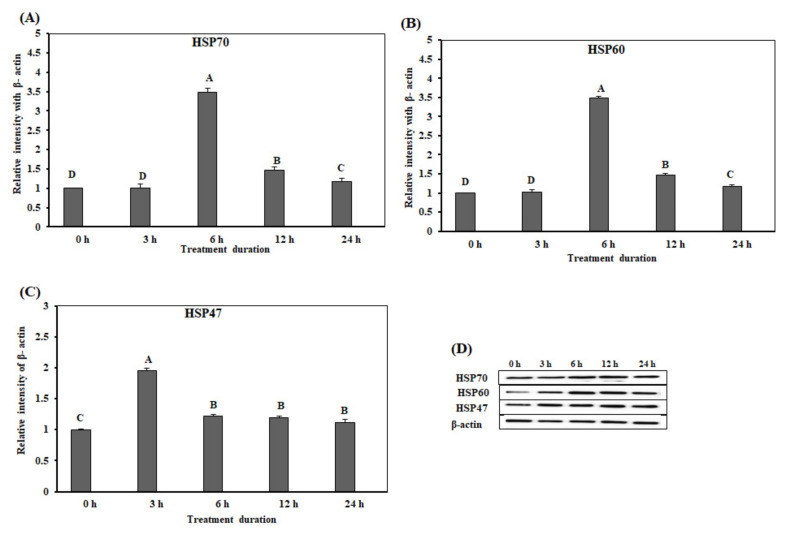
The expression of the heat shock proteins in the duodenum of chickens exposed to acute heat stress for different durations of time (*n* = 8). The mean relative expression of *HSP70* (**A**), *HSP60* (**B**), and *HSP47* (**C**). (**D**) Western blot analysis results. ^A–D^
*HSP70*, *HSP60*, and *HSP47* in the columns with the different superscript letters differ significantly (*p* < 0.05) among the different heat exposure durations.

**Figure 5 animals-10-01234-f005:**
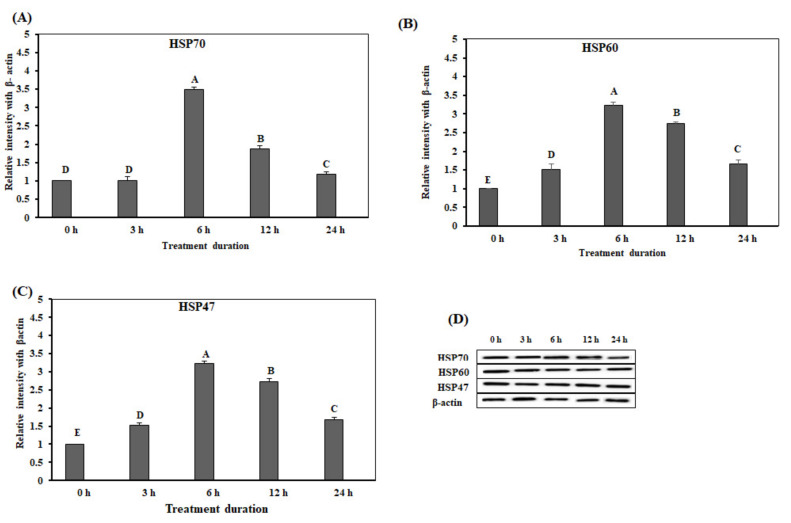
The protein expression of the heat shock proteins in the jejunum of chickens exposed to acute heat stress for different durations of time (*n* = 8). The mean relative expression of *HSP70* (**A**), *HSP60* (**B**), and *HSP47* (**C**). (**D**) Western blot analysis results. ^A–E^
*HSP70*, *HSP60*, and *HSP47* in the columns with the different superscript letters differ significantly (*p* < 0.05) among the different heat exposure durations.

**Figure 6 animals-10-01234-f006:**
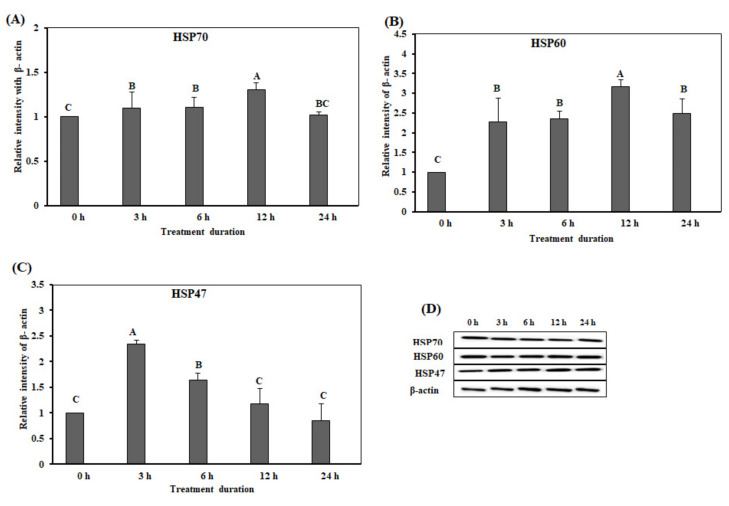
The protein expression of the heat shock proteins in the ileum of chickens exposed to acute heat stress for different durations of time (*n* = 8). The mean relative expression of *HSP70* (**A**), *HSP60* (**B**), and *HSP47* (**C**). (**D**) Western blot analysis results. ^A–C^
*HSP70*, *HSP60*, and *HSP47* in the columns with the different superscript letters differ significantly (*p* < 0.05) among the different heat exposure durations.

**Figure 7 animals-10-01234-f007:**
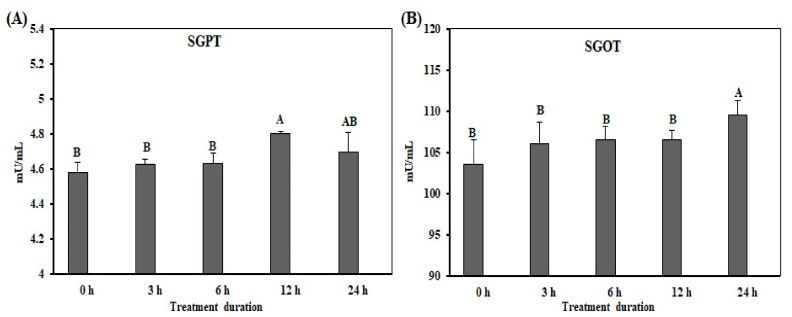
Effect of acute heat stress on the levels of serum glutamate pyruvate transaminase (SGPT; (**A**)) and serum glutamate oxaloacetate transaminase (SGOT; (**B**)) in chickens (*n* = 8). ^A,B^ SGPT and SGOT in the columns with the different superscript letters differ significantly (*p* < 0.05) among the different heat exposure durations.

**Table 1 animals-10-01234-t001:** Primers used for qRT–PCR.

Primer Name	Primer Sequence (5′→ 3′)	Annealing Temperature (°C)
*HSP70*	F-GGTAAGCACAAGCGTGACAATGCT	55
	R-TCAATCTCAATGCTGGCTTGCGTG	
*HSP60*	F-AGAAGAAGGACAGAGTTACC	55
	R-GCGTCTAATGCTGGAATG	
*HSP47*	F-ACTGGCTCATAAGCTCTCCAGCAT	57
	R-TCATCTTGCTGGCCCA GGTCTTTA	
*GAPDH*	F-AGAACATCATCCCAGCGTCC	60
	R-CGGCAGGTCAGGTCAACAAC	

HSP; heat shock protein, *GAPDH*; Glyceraldehyde 3-phosphate dehydrogenase

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
