# Peer review of "Acute Heat Stress Induces the Differential Expression of Heat Shock Proteins in Different Sections of the Small Intestine of Chickens Based on Exposure Duration"

_animals, 2020, doi:10.3390/ani10071234_

Round 1

Reviewer 1 Report

Comments attached!

Author Response

Kwanseob Shim, PhD

                                                                   Dept of Animal Biotechnology

Jeonbuk National University

Jeonju, 54896.

Republic of Korea

Vladislava Sesum,
Animals Editorial Office
MDPI Novi Sad Office
Arse Teodorovica 15, 21000 Novi Sad, Serbia

Dear Vladislava Sesum,

The enclosed manuscript is our revised manuscript entitled “Acute heat stress induces the differential expression of heat shock proteins in different sections of the small intestine of chickens based on exposure duration” by S H Siddiqui et al for publication in Animals (Manuscript number: animals-847046).We appreciate the time and efforts by the editor and reviewer of Animals in reviewing our manuscript. We corrected and answered all of points addressed by the reviews, and believed that the revised version can meet the publication requirements of Animals. We are deeply thankful for your helpful comments and suggestions. These comments have helped us a lot in clarifying and sharpening our manuscript. Following your suggestion, we have modified the manuscript accordingly, and detailed corrections are listed as below.

Point-by-point response to Reviewer #1

COMMENT#1:

Line 33: Instead of …early treatment group… it would be better to write…during 3h heat exposure…

Response to Comment #1:

We would like to convey our thanks to the reviewer for his valuable suggestion. Based on reviewer’s suggestion, we have rewrite “3 h treatment group” instead of “early treatment group”. Please refer to line 33 for the change.

COMMENT#2:

Line 36: What can be understood under …”indication of heat tolerance”? It has to be explained with a couple of word in summary.

Response to Comment #2:

As per reviewer’s query, we explained the recommended term in conclusion section. Please refer to line 309-310 for the change.

COMMENT#3:

Line 59: It is emphasized again: English in high standards avoid starting a sentence with abbreviation or number.

Response to Comment #3:

We would like to convey our thanks to the reviewer for his valuable suggestion. Following reviewer’s suggestion, we have change the starting of recommended sentence. Please refer to line 59 for the change.

COMMENT#4:

Line 75: Additionally, damage of liver tissues was tried to indicated by measuring SGPD and SGOT level in blood serum.

Response to Comment #4:

Following the reviewer’s remark, we re-write the sentence “Additionally, damage to liver tissues was tried to indicate by measuring SGPT and SGOT levels in blood serum” instead of “Additionally, we examined the effect of acute heat stress on the serum levels of SGPT and SGOT”. Please refer to lines 77-78 for the change.

COMMENT#5:

Line 89: In total, 128 chickens aged 21 days, weighing…

Response to Comment #5:

As per the reviewer’s remarks, we re-write the recommended sentence “In total, 128 chickens aged 21 days”. Please refer to line 89-90 for the change.

COMMENT#6:

Lines 92-95: Moisture content of the air or Temperature Humidity Index (THI) indication during heat stress should be useful.

Response to Comment #6:

Based on reviewer’s suggestion, we added relative humidity of treatment groups in materials and method section. Please refer to line 95-96 for the change.

COMMENT#7:

Lines 164-166: This sentence belongs to methods, and already written in lines 114-115; also in lines 188 and 226 sentences should be omitted.

Response to Comment #7:

Following the reviewer’s suggestion, we omitted the recommended sentences. Please refer to lines 165, 186, and 223 for the changes.

Afterward, we appreciate your valuable suggestions. Please let me know if we did not answer all of your comments clearly or if you require any further information.

Best regards

Kwanseob Shim, PhD

Reviewer 2 Report

The manuscript "Acute heat stress induces differential expression of heat shock proteins in different sections of the chicken small intestine based on exposure duration " has an appropriate methodology and describes important results on the acute heat stress in chickens.

The authors should correct the text "in the jejunum on acute heat stress 182 (0, 3, 6, 12, 24 h) were significantly higher than that in the control group (p < 0.05)" to "in the jejunum on acute heat stress 182 (3, 6, 12, 24 h) were significantly higher than that in the control group (p < 0.05)"

Author Response

Kwanseob Shim, PhD

                                                                   Dept of Animal Biotechnology

Jeonbuk National University

Jeonju, 54896.

Republic of Korea

Vladislava Sesum,
Animals Editorial Office
MDPI Novi Sad Office
Arse Teodorovica 15, 21000 Novi Sad, Serbia

Dear Vladislava Sesum,

The enclosed manuscript is our revised manuscript entitled “Acute heat stress induces the differential expression of heat shock proteins in different sections of the small intestine of chickens based on exposure duration” by S H Siddiqui et al for publication in Animals (Manuscript number: animals-847046).We appreciate the time and efforts by the editor and reviewer of Animals in reviewing our manuscript. We corrected and answered all of points addressed by the reviews, and believed that the revised version can meet the publication requirements of Animals. We are deeply thankful for your helpful comments and suggestions. These comments have helped us a lot in clarifying and sharpening our manuscript. Following your suggestion, we have modified the manuscript accordingly, and detailed corrections are listed as below.

Point-by-point response to Reviewer #2

COMMENT# 1:

The authors should correct the text "in the jejunum on acute heat stress 182 (0, 3, 6, 12, 24 h) were significantly higher than that in the control group (p < 0.05)" to "in the jejunum on acute heat stress 182 (3, 6, 12, 24 h) were significantly higher than that in the control group (p < 0.05)"

Response to Comment #1

First of all, we would like to thank reviewer for his keen observation. As per reviewer’s remark, we have omitted 0 h from the recommended line. Please refer line 180 for change.

Afterward, we appreciate your valuable suggestions. Please let me know if we did not answer all of your comments clearly or if you require any further information.

Best regards

Kwanseob Shim, PhD

Reviewer 3 Report

The paper presented for review brings new elements to the current state of knowledge regarding the impact of acute heat stress on protein expression in various sections of the small intestine in broiler chickens.

The purpose of the work is clearly stated. The conclusions of the conducted research are clear and result from the obtained research results.

The material used for the tests is sufficient, the test methods have been selected accordingly.

The figures correctly present the results obtained. The differences between the groups were marked correctly.

Discussing the results against the background of other authors is very detailed. The publications cited by the authors of the article are well selected. For the most part, the authors refer to the latest knowledge published in renowned scientific journals.

Author Response

Kwanseob Shim, PhD

                                                                   Dept of Animal Biotechnology

Jeonbuk National University

Jeonju, 54896.

Republic of Korea

Vladislava Sesum,
Animals Editorial Office
MDPI Novi Sad Office
Arse Teodorovica 15, 21000 Novi Sad, Serbia

Dear Vladislava Sesum,

The enclosed manuscript is our revised manuscript entitled “Acute heat stress induces the differential expression of heat shock proteins in different sections of the small intestine of chickens based on exposure duration” by S H Siddiqui et al for publication in Animals (Manuscript number: animals-847046).We appreciate the time and efforts by the editor and reviewer of Animals in reviewing our manuscript. We corrected and answered all of points addressed by the reviews, and believed that the revised version can meet the publication requirements of Animals. We are deeply thankful for your helpful comments and suggestions. These comments have helped us a lot in clarifying and sharpening our manuscript. Following your suggestion, we have modified the manuscript accordingly, and detailed corrections are listed as below.

Point-by-point response to Reviewer #3

COMMENT# 1: 

The paper presented for review brings new elements to the current state of knowledge regarding the impact of acute heat stress on protein expression in various sections of the small intestine in broiler chickens.

The purpose of the work is clearly stated. The conclusions of the conducted research are clear and result from the obtained research results.

The material used for the tests is sufficient, the test methods have been selected accordingly.

The figures correctly present the results obtained. The differences between the groups were marked correctly.

Discussing the results against the background of other authors is very detailed. The publications cited by the authors of the article are well selected. For the most part, the authors refer to the latest knowledge published in renowned scientific journals.

Response to Comment #1:

First of all, we would like to express our gratitude to the reviewer for reviewing our manuscript thoroughly. These compliments of the reviewer will encourage us in our future research. We are grateful to you for your nice compliments.

Afterward, we appreciate your valuable suggestions. Please let me know if we did not answer all of your comments clearly or if you require any further information.

Best regards

Kwanseob Shim, PhD

This manuscript is a resubmission of an earlier submission. The following is a list of the peer review reports and author responses from that submission.

Round 1

Reviewer 1 Report

The paper presented for review brings new elements to the current state of knowledge regarding the impact of acute heat stress on protein expression in various sections of the small intestine in broiler chickens.

The purpose of the work is clearly stated. The conclusions of the conducted research are clear and result from the obtained research results.

The material used for the tests is sufficient, the test methods have been selected accordingly. The methodology should include the following information:

  1. Were experimental groups exposed to high temperature only once on day 21 of breeding or were they subjected to high temperature from day 21 until the end of breeding?
  2. How long did chick rearing take place and at what age the material for testing was collected from chickens?

The figures correctly present the results obtained. The differences between the groups were marked correctly.

Discussing the results against the background of other authors is very detailed. The publications cited by the authors of the article are well selected. For the most part, the authors refer to the latest knowledge published in renowned scientific journals.

Author Response

Kwanseob Shim, PhD

                                                      Dept of Animal Biotechnology

Jeonbuk National University

Jeonju, Chonbuk 54896.

Republic of Korea

Vladislava Sesum,
Animals Editorial Office
MDPI Novi Sad Office
Arse Teodorovica 15, 21000 Novi Sad, Serbia

Dear Vladislava Sesum,

The enclosed manuscript is our revised manuscript entitled “Acute heat stress induces differential expression of heat shock proteins in different sections of the chicken small intestine based on exposure duration” by S H Siddiqui et al for publication in Animals (Manuscript number: animals-778500).

We appreciate the time and efforts by the editor and reviewer of Animals in reviewing our manuscript. We corrected and answered all of points addressed by the reviews, and believed that the revised version can meet the publication requirements of Animals. We are deeply thankful for your helpful comments and suggestions. These comments have helped us a lot in clarifying and sharpening our manuscript. Following your suggestion, we have modified the manuscript accordingly, and detailed corrections are listed as below.

Point-by-point response to Reviewer #1

COMMENT#1:

Were experimental groups exposed to high temperature only once on day 21 of breeding or were they subjected to high temperature from day 21 until the end of breeding?

Response to Comment #1:

First of all, we would like to thank reviewer to consider for publication of our study. As per reviewer’s query, we explained that experimental groups were exposed to high temperature only once on day 21 for 24-h. Please refer line 95-96 for the change.

COMMENT#2:

How long did chick rearing take place and at what age the material for testing was collected from chickens?

Response to Comment #2:

As per reviewer’s remark, we explained that chick rearing took place for 21 days and the material for testing was collected from chickens on 21 days at 0-h, 3-h, 6-h, 12-h and 24-h. We appreciate your valuable suggestion.

Afterwards, we appreciate your valuable suggestions. Please let me know if we did not answer all of your comments clearly or if you require any further information.

Best regards

Kwanseob Shim, PhD

Reviewer 2 Report

The manuscript "Acute heat stress induces differential expression of heat shock proteins in different sections of the chicken small
intestin based on exposure duration" is about a relevant topic, has an appropriate methodology and describes important results on the acute heat stress in chickens.
However, there are issues that need to be clarified by the authors.
1) The authors state that there was intestinal injury, but there is no evidence of this in the results. The authors
show increased levels of SGPT and SGOT that are markers of liver damage but not intestinal damage. The authors could have used other methodologies such as histochemistry to show these lesions.
2) The methodology does not adequately describe how many animals were used in each experiment.
The authors report "The treatment group were subdivided into 4 subgroups based on the duration of exposure (3-h, 6-h, 12-h, and 24-h) where control group has 0-h of exposure. The chickens were maintained in a cage (15 birds per cage) and provided feed and water ad libitum. Each group had 8 cages (8 replications) for each treatment group. "
So 8 cages x 15 birds per cage = 120 birds at 3-h, 120 birds at 6-h ... Is that correct?
Authors must inform how many animals were used in qPCR and Western blotting analyzes.
3) The authors used β-actin and GAPDH as normalized for protein and gene expression. However, Lin & Redies 2012* showed that there is differences in the expression of these markers depending on the tissue analyzed and this can directly influence the results presented in this study.
Therefore, the authors should carry out further analyzes using anti-GAPDH antibodies in Western blotting and primers for β-actin in qPCR for normalization of protein and gene expression.
*Lin J, Redies C. Histological evidence: housekeeping genes beta-actin and GAPDH are of limited value for normalization of gene expression. Dev Genes Evol.  2012 Nov;222(6):369-76. doi: 10.1007/s00427-012-0420-x.

Author Response

Kwanseob Shim, PhD

                                                            Dept of Animal Biotechnology

Jeonbuk National University

Jeonju, Chonbuk 54896.

Republic of Korea

Vladislava Sesum,
Animals Editorial Office
MDPI Novi Sad Office
Arse Teodorovica 15, 21000 Novi Sad, Serbia

Dear Vladislava Sesum,

The enclosed manuscript is our revised manuscript entitled “Acute heat stress induces differential expression of heat shock proteins in different sections of the chicken small intestine based on exposure duration” by S H Siddiqui et al for publication in Animals (Manuscript number: animals-778500).

We appreciate the time and efforts by the editor and reviewer of Animals in reviewing our manuscript. We corrected and answered all of points addressed by the reviews, and believed that the revised version can meet the publication requirements of Animals. We are deeply thankful for your helpful comments and suggestions. These comments have helped us a lot in clarifying and sharpening our manuscript. Following your suggestion, we have modified the manuscript accordingly, and detailed corrections are listed as below.

Point-by-point response to Reviewer #2

COMMENT# 1:

The authors state that there was intestinal injury, but there is no evidence of this in the results. The authors show increased levels of SGPT and SGOT that are markers of liver damage but not intestinal damage. The authors could have used other methodologies such as histochemistry to show these lesions?

Response to Comment #1

As per reviewer’s remark, we would like to clarify that we never mentioned about intestinal injury. We explained a relation between liver function and intestinal molecular structure. Please refer line 259 and 260 for the changes.

COMMENT# 2:

The methodology does not adequately describe how many animals were used in each experiment. The authors report "The treatment group was subdivided into 4 subgroups based on the duration of exposure (3-h, 6-h, 12-h, and 24-h) where control group has 0-h of exposure. The chickens were maintained in a cage (15 birds per cage) and provided feed and water ad libitum. Each group had 8 cages (8 replications) for each treatment group. " So 8 cages x 15 birds per cage = 120 birds at 3-h, 120 birds at 6-h ... Is that correct? Authors must inform how many animals were used in qPCR and Western blotting analyzes.

Response to Comment #2:

No. 120 birds at 3-h is not correct. We wrote some part of methodology. Please refer line 92-93, 100-101 and 107-108 for the changes. We appreciate your valuable suggestions.

COMMENT# 3:

The authors used β-actin and GAPDH as normalized for protein and gene expression. However, Lin & Redies 2012* showed that there is differences in the expression of these markers depending on the tissue analyzed and this can directly influence the results presented in this study. Therefore, the authors should carry out further analyzes using anti-GAPDH antibodies in Western blotting and primers for β-actin in qPCR for normalization of protein and gene expression.

*Lin J, Redies C. Histological evidence: housekeeping genes beta-actin and GAPDH are of limited value for normalization of gene expression. Dev Genes Evol.2012 Nov;222(6):369-76. doi: 10.1007/s00427-012-0420-x.

Response to Comment #3:

We respect reviewer’s suggestion. However, it has been reported that GAPDH gene expression level in qPCR are comparatively high in liver and heart muscle (Lin and Redies 2012). In addition, previous studies have used GAPDH as a housekeeping gene in intestine (Doyun et al., 2019; Xiaofang et al., 2017; Huang S., 2017). On the other hand, β-actin has been widely used in western blot in tissue extract (Rufang et al., 2012). Moreover, previous studies have used β-actin as an internal control in western blot analysis (Sivakumar et al., 2020; Soheil et al., 2015). Due to above mentioned references, we used β-actin and GAPDH for normalized the protein and gene expression, respectively in our study.

Afterwards, we appreciate your valuable suggestions. Please let me know if we did not answer all of your comments clearly or if you require any further information.

Best regards

Kwanseob Shim, PhD

Reviewer 3 Report

It is very interesting research that indicated the heat shock protein expressions are different in each part of the intestine and during various heat stress duration. However, the primers used in the current research were problematic. The writing of this manuscript, especially the indication of results, is not precise and needs significant improvement. The discussion did not relate to the key findings; at last, there are some concerns in the statistics.
The major issues of the current manuscript are:

1. primers: HSP60 sequence was checked, which is HSP 90 instead of 60. GAPDH sequence has O in it, which should be C.

2.statistic: The control group (0h) has 120 birds. My understanding is whenever the sampling happened (3, 6, 12 or 24h), 8 birds will be sampled from control as well. In this case, the author should have at least 4 sets of control birds. However, in the results, only one control was shown in the figures. What is this control?

3. the interpretation of the results needs to be improved significantly. The details of how these proteins/mRNA been regulated under different heat stress time need to be described in detail. The summary right after each section needs to be more precise.
4. the discussion did not relate to the key results. The detailed comparison of the current study to the previous research needs to be discussed in the content. When talking about the results, the author needs to make it clear that the results are referring to heat shock protein expression or mRNA expression.
5. The conclusion is too rough. These differences need to be summarized. The results may help the researcher decided when and where should be sampled in a heat stress trial. According to the results, the author can provide suggestions.

Details:
Line 35. Liver?
Line 108. Is it an ELISA kit?
Line 163, at 12 h or 24 h compared to the other group, respectively. Similar errors were seen throughout the manuscript.
Line 175: significantlyb?
Line 179: such a summary is not acceptable.
Line 189: very confusing. Did it just repeat the previous sentence?
Figure 3D and 4D are not mentioned in the manuscript. The image quality is not good enough to see any difference.
Line 200: confusing summary.
Line 299: the expression levels are not comparable, but the trend is. We need more discussion in the relationship between RNA and protein levels. Or indicate what the point of testing both is.

Author Response

(The authors gave the same response as above.)

Reviewer 4 Report

File is attached

Author Response

Kwanseob Shim, PhD

                                                            Dept of Animal Biotechnology

Jeonbuk National University

Jeonju, Chonbuk 54896.

Republic of Korea

Vladislava Sesum,
Animals Editorial Office
MDPI Novi Sad Office
Arse Teodorovica 15, 21000 Novi Sad, Serbia

Dear Vladislava Sesum,

The enclosed manuscript is our revised manuscript entitled “Acute heat stress induces differential expression of heat shock proteins in different sections of the chicken small intestine based on exposure duration” by S H Siddiqui et al for publication in Animals (Manuscript number: animals-778500).

We appreciate the time and efforts by the editor and reviewer of Animals in reviewing our manuscript. We corrected and answered all of points addressed by the reviews, and believed that the revised version can meet the publication requirements of Animals. We are deeply thankful for your helpful comments and suggestions. These comments have helped us a lot in clarifying and sharpening our manuscript. Following your suggestion, we have modified the manuscript accordingly, and detailed corrections are listed as below.

Point-by-point response to Reviewer #4

COMMENT# 1: 

Line 12: …organ in the alimentary tract of chicken

Response to Comment #1:

We are grateful to you for your kind suggestion. Based on reviewer’s suggestion we revised the sentence from “Small intestine is a pivotal organ of chicken where around 90% digestion and absorption occurs” to “Small intestine is a pivotal organ in the alimentary tract of chicken where around 90% digestions and absorption occurs”. Please refer line 12 for the change.

COMMENT# 2:

Lines 12-14: Written lines (first 2 sentences can be omitted)

Response to Comment #2:

We would like to convey our thanks to the reviewer for his valuable suggestion. However, we rewrote these sentences. Please refer line 12-13 for the change.

COMMENT# 3:

Line 14: Previous studies…

Response to Comment #3:

Following reviewer’s remark, we corrected this sentence. Please refer line 13 for the change.

COMMENT# 4:

Lines 22-35 (Summary): This summary does not shows clearly the essence of the study. At the beginning of this part one or two sentences could be composed describing the main points of the experimental protocol, including the specifications of animals (breed, age) and treatment groups. After showing the experimental protocol, at first the results indicating HSPs should be described, then SGOT and SGPT changes can be described. Later in the “Result” it also will be mentioned.

Instead of “glutamic pyruvic and glutamic oxalacetic transaminase” it is better to write “glutamate pyruvate and glutamate oxalacetate transaminase”

Response to Comment #4:

Based on reviewer’s suggestion, we added aim and experimental protocol, rearranged the result sequence and changed the elaboration of SGPT and SGOT. Please refer line 21-24, 33 for the change.

COMMENT# 5:

Lines 34-35: ...stress affects what parameters in intestine and liver…?

Response to Comment #5:

As per reviewer’s remarks, we fixed the parameters. Please refer line 35 for the change.

COMMENT# 6:

Lines 40: In tropical countries and even in mild climate areas, as a consequence of climate changes, the poultry industry…

Response to Comment #6:

Following reviewer’s suggestion, we added these words for improving the sentence. Please refer line 41-42 for the change. We appreciate your valuable suggestions.

COMMENT# 7:

Lines 44: ... organs or tissues.

Response to Comment #7:

As per reviewer’s comments, we fixed these words. Please refer line 45- 46 for the change.

COMMENT# 8:

Lines 49: …One of the indicators of this heat stress…

Response to Comment #8:

Following reviewer’s suggestion, we revised the sentence. Please refer the line 51 for the change.

COMMENT# 9:

Line 51 and later everywhere: Never start a sentence with abbreviation or numbers! “Heat stress proteins”or “The HSPs”…

Response to Comment #9:

As per reviewer’s suggestion, we revised the manuscript. We are grateful to you for your valuable suggestion.

COMMENT# 10:

Line 62: …which provides” (Omit word” it”!)

Response to Comment #10:

Based on reviewer’s recommendation, we omitted the word ‘it’ from the sentence. Please refer line 65 for the change.

COMMENT# 11:

Line 66: …, which provides substances for growth and development…

Response to Comment #11:

As per reviewer’s comments, we rewrite the sentence. Please refer line 69 for the change.

COMMENT# 12:

Lines 72-73: instead of…based on the duration” … correct form: during acute heat stress…

Response to Comment #12:

As per reviewer’s remarks, we revised the recommended line. Prefer line 75-76 for the change.

COMMENT# 13:

line 81: …protocol, and were approved by…

Response to Comment #13:

Following reviewer’s suggestion, we revised the sentence. Please refer line 84 for the change.

COMMENT# 14:

Line 88: …per week, and temperature was fixed at 26±1°C, and…

Response to Comment #14:

As per reviewer’s comments, we corrected the sentence. Please refer line 91-92 for change.

COMMENT# 15:

Line 90: Chickens of control group were kept at 26±1°C and…

Response to Comment #15:

Following reviewer’s comments, we fixed the recommended sentence. Please refer line 94 for the change.

COMMENT# 16:

Lines 92-93: …based on the duration of heat exposure…; Chickens of control group had 0h of heat exposure…

Response to Comment #16:

Based on reviewer’s comments, we revised the sentence. Please refer line 96-97 for the change.

COMMENT# 17:

Line 94: … ad libitum throughout the experimental period…

Response to Comment #17:

As per reviewer’s comments, we rewrite the sentence. Please refer line 98 for the change.

COMMENT# 18:

Lines 95-96: Last sentence of this chapter has to be omitted, because it is written earlier!

Response to Comment #18:

Based on your in-depth suggestion, we omitted the repeated sentence. Please refer line 98 for the change.

COMMENT# 19:

Lines 98-103: What were the dividing lines between different intestinal parts during dissection? It is important to indicate, because there are two different definitions for the dividing lines between jejunum and ileum.

Response to Comment #19:

As per reviewer’s comments, we indicated the separating line of different section of small intestine. Please refer line 103-105 for the change.

COMMENT# 20:

Lines 111 or somewhere here: Which parts and what weights of different intestinal sections were collected for analyses?

Response to Comment #20:

As per reviewer’s remarks, we added name of intestinal three different sections and weight of sample, which were collected for analyses. Please refer line 119 for the change.

COMMENT# 21:

Lines 159-162: First two sentences include of this chapter have to be omitted. In the chapter “Results” should be included only the description of results. Methods and discussion parts have to be omitted from here. Describing the results past tense should be used instead of present.

Response to Comment #21:

Following reviewer’s suggestion, we omitted that two sentence and we changed the tense. Please refer line 238-240 for the change.

COMMENT# 22:

General remark: The main aim of this study is to show the effect of different heat stress on HPSs as it also reflected even in the title of the manuscript. This way, it is better to describe or discusses at first the results relating to HSPs, then blood enzyme changes can be followed. For this reason, the order of chapters 3.1 and 3.3 has to be changed. In other chapters the same instructions could be followed.

Response to Comment #22:

As per reviewer’s suggestion, we rearranged 3.1 and 3.3 chapter between themselves. We are grateful to you for your valuable suggestion.

COMMENT# 23:

Lines 155: …among groups exposed to heat stress for 0 h, 3 h, 6 h and 24 h, respectively.

Response to Comment #23:

As per reviewer’s recommendation, we rewrite the sentence. Please refer line 164 for the change.

COMMENT# 24:

Line 163: …higher measured at 12 h and 24 h heat exposure…

Response to Comment #24:

As per reviewer’s comments, we revised the sentence. Please refer line 240 for the change.

COMMENT# 25:

Lines 164-165: The last sentence of this part belongs to chapter “Discussion”.

Response to Comment #25:

Based on the reviewer’s remark, we omitted the sentence from result parts. Please refer the line 240 for the change.

COMMENT# 26:

Title of Fig. 1: Effect of acute heat stress on the serum glutamate pyruvate transaminase (SGPT) and serum glutamate oxaloacetate transaminase (SGOT) in chickens. A-B SGPT and…

Response to Comment #26:

Based on reviewer’s comments, we changed the elaboration of SGPT and SGOT. Please refer line 242-243 for the change.

COMMENT# 27:

Lines 172-174: These two sentences should be included in the chapter “Materials and methods”.

Response to Comment #27:

As per reviewer’s comments, we shifted two sentence from ‘results’ part to “Materials and methods” section. Please refer line 116-118, 202 for the change.

COMMENT# 28:

Line 178: …treatment groups

Response to Comment #28:

Following reviewer’s comments, we revised the sentence. Please refer line 205 for the change.

COMMENT# 29:

Line 179: Overall, each of the three HSPs expression level was significantly higher, respectively compared to…

Response to Comment #29:

Based on reviewer’s remark, we replaced the sentence from “Overall, three HSPs expression levels were significantly higher compared to control group (0 h)” with “Overall, each of the three HSPs expression level of treatment groups were significantly higher, respectively compared to 0-h.” Please refer line 206-208 for the change.

COMMENT# 30:

Line 201: Sentence is closed by 2 points. Remove one of them!

Response to Comment #30:

Following reviewer’s remark, we removed one points. Please refer line 230 for the change.

COMMENT# 31:

Lines 222: word “showed”: present tense instead of past

Response to Comment #31:

As per reviewer’s remark, we replaced the word “showed” with “represents”. Please refer line 181 for the change.

COMMENT# 32:

Line 223: ...were significantly higher (P<0.05) in the four heat stressed groups than in control.

Response to Comment #32:

Following reviewer’s suggestion, we rewrote the sentence. Please refer line 186 for the change.

COMMENT# 33:

Lines 224-225: In the text, description of the results shown in Table 6. is not accurate, enough. …gene expression level was significantly higher at 12 h and6 h, respectively, than which group(s)? At the same time 3 h and 24 h heat exposure resulted also in significant differences comparing to control.

Table 6 also shows differences among different time durations of heat exposure. This remark can be also added to almost all descriptions in the text of results shown in the tables.

Response to Comment #33:

As per reviewer’s remark, we rewrote the result section. Please refer line 181-186 for the change.

COMMENT# 34:

Higher activity changes of the two blood enzymes tested in the experiment show damages not only the liver cells, but other somatic cells, mostly in heart and kidney. This way, the activity changes of these enzymes have only slight indications on liver cell damages in heat stress. At heat stress there are more precious measurements indicating liver cell damages (Mackei et al., 2020: Effects of acute heat stress on a newly established chicken hepatocyte, Animals, Vol: 10, Issue: 3). For this reason it would be very useful to find any connections between the two enzymes functions and HSPs in the tissues of chickens. It would increase the value of discussion part of the manuscript.

Almost all three HSPs expressions in the different intestinal sections increased after heat exposure, peaked at 6-12 duration of exposure, and then decreased significantly. However, the values never decreased under the control. This way, the statement that the HSPs expressions declined at the end of the experiment, compared to control is not reflected from the data of figures.

More detailed discussion would be required for the reason of trends that the expression of HSPs increased up to 6-12 hours of heat stress treatment, and then declined significantly, but never declined under the control values.

Response to Comment #34:

Following reviewer’s suggestion, we added the detailed discussion about HSPs increased up to 6-12 hours and relation of injured liver and intestine. We added line 259-260 and 309-312 in the manuscript. We appreciate your valuable suggestion.

Afterwards, we appreciate your valuable suggestions. Please let me know if we did not answer all of your comments clearly or if you require any further information.

Best regards

Kwanseob Shim, PhD

Round 2

Reviewer 2 Report

The revised manuscript entitled “Acute heat stress induces differential expression of heat shock proteins in different sections of the chicken small intestine based on exposure duration” is suitable for publication.

Minor corrections:

lines 206, 216 e 230 – westernblotting to western blotting.

Author Response

Kwanseob Shim, PhD

                                                                   Dept of Animal Biotechnology

Jeonbuk National University

Jeonju, Chonbuk 54896.

Republic of Korea

Vladislava Sesum,
Animals Editorial Office
MDPI Novi Sad Office
Arse Teodorovica 15, 21000 Novi Sad, Serbia

Dear Vladislava Sesum,

The enclosed manuscript is our revised manuscript entitled “Acute heat stress induces differential expression of heat shock proteins in different sections of the chicken small intestine based on exposure duration” by S H Siddiqui et al for publication in Animals (Manuscript number: animals-778500).We appreciate the time and efforts by the editor and reviewer of Animals in reviewing our manuscript. We corrected and answered all of points addressed by the reviews, and believed that the revised version can meet the publication requirements of Animals. We are deeply thankful for your helpful comments and suggestions. These comments have helped us a lot in clarifying and sharpening our manuscript. Following your suggestion, we have modified the manuscript accordingly, and detailed corrections are listed as below.

Point-by-point response to Reviewer #2

COMMENT# 1:

lines 206, 216 and 230 – westernblotting to western blotting.

Response to Comment #1

First of all, we would like to thank reviewer to consider for publication of our study. As per reviewer’s remark, we replaced “westernblotting” with “western blotting”. Please refer line 208, 218 and 232 for the change. We appreciate your valuable suggestion.

Afterwards, we appreciate your valuable suggestions. Please let me know if we did not answer all of your comments clearly or if you require any further information.

Best regards

Kwanseob Shim, PhD

Reviewer 3 Report

Please check the attachment for detail. The top concern is the primer is not right. 

Author Response

Kwanseob Shim, PhD

                                                                   Dept of Animal Biotechnology

Jeonbuk National University

Jeonju, Chonbuk 54896.

Republic of Korea

Vladislava Sesum,
Animals Editorial Office
MDPI Novi Sad Office
Arse Teodorovica 15, 21000 Novi Sad, Serbia

Dear Vladislava Sesum,

The enclosed manuscript is our revised manuscript entitled “Acute heat stress induces differential expression of heat shock proteins in different sections of the chicken small intestine based on exposure duration” by S H Siddiqui et al for publication in Animals (Manuscript number: animals-778500).We appreciate the time and efforts by the editor and reviewer of Animals in reviewing our manuscript. We corrected and answered all of points addressed by the reviews, and believed that the revised version can meet the publication requirements of Animals. We are deeply thankful for your helpful comments and suggestions. These comments have helped us a lot in clarifying and sharpening our manuscript. Following your suggestion, we have modified the manuscript accordingly, and detailed corrections are listed as below.

Point-by-point response to Reviewer #3

COMMENT# 1: 

Please see the blast details below. I don’t think it is HSP60. If so, please explain. Otherwise, the manuscript need to be changed accordingly.

Products on target templates

>NM_204289.1 Gallus gallus heat shock protein 90 beta family member 1 (HSP90B1), mRNA

product length = 172

Forward primer 1 ATGTGTGGAGCAGCAAGACAGAGA 24

Template 937 ........................ 960

Reverse primer 1 TTCATGAGCTCCCAATCCCAGACA 24

Template 1108 ........................ 1085

Response to Comment #1:

We apologize for the mistake. We have corrected the primer sequence of HSP60. Please refer line 160 (table 1) for the change.  We have also performed the experiment with the corrected primer sequence of HSP60 to observe HSP60 mRNA expression. We have added the result of HSP60 mRNA expression. Please refer line 176-177, 185-186 and 195-196 for the change. We are grateful to you for your keen observation.  

COMMENT# 2:

I did not get the idea of how the study was designed. my understanding is 2 group of bird were set. 120 birds were only for control without any heat treatment. Another 120 birds were under heat stress for 24 hours. At 3, 6, 12, 24h, the 8 birds from the second group were sampled. However, how the sampling goes in control group? If you only need 8 birds from control (out of 120), why don’t you take 8 birds before heat stress, then sample from same group during the heat stress. Which will save half number of birds without effecting the confidence of data.

Otherwise, with the sampling going in heat stress group, were control group also sampled at 3h, 6h, 12h and 24hr? then you should have multiple control. Just present one of them is not right.

Please be clear on experiment design and make sure no experimental animals are sacrificed but no contributing to the research.

Response to Comment #2:

Following reviewer’s suggestion, we have corrected the experimental design. Please refer line 92-100 for the change. We appreciate your valuable suggestion.

COMMENT#3: 

Please describe the kits, it is hard to find what exactly kits is by the information provided.

Response to Comment #3

As per reviewer’s query, we added the name of kit type. Please refer line 115 for the change. We appreciate your valuable suggestion.

Afterwards, we appreciate your valuable suggestions. Please let me know if we did not answer all of your comments clearly or if you require any further information.

Best regards

Kwanseob Shim, PhD

Reviewer 4 Report

All the suggestions of the reviewer were accepted by the authors, and they corrected the manuscript well. This way the merit of the article improved.

The reviewer suggest two more language corrections in the manuscript as follows:

line 35: ...affect intestinal proteins… (word "in can be omitted)

line t2: ...is the group of heat shock proteins…(word "One is single", "proteins is plural) 

Author Response

Kwanseob Shim, PhD

                                                                   Dept of Animal Biotechnology

Jeonbuk National University

Jeonju, Chonbuk 54896.

Republic of Korea

Vladislava Sesum,
Animals Editorial Office
MDPI Novi Sad Office
Arse Teodorovica 15, 21000 Novi Sad, Serbia

Dear Vladislava Sesum,

The enclosed manuscript is our revised manuscript entitled “Acute heat stress induces differential expression of heat shock proteins in different sections of the chicken small intestine based on exposure duration” by S H Siddiqui et al for publication in Animals (Manuscript number: animals-778500).We appreciate the time and efforts by the editor and reviewer of Animals in reviewing our manuscript. We corrected and answered all of points addressed by the reviews, and believed that the revised version can meet the publication requirements of Animals. We are deeply thankful for your helpful comments and suggestions. These comments have helped us a lot in clarifying and sharpening our manuscript. Following your suggestion, we have modified the manuscript accordingly, and detailed corrections are listed as below.

Point-by-point response to Reviewer #4

COMMENT# 1: 

Line 35: affect intestinal proteins… (word "in can be omitted)

Response to Comment #1:

Based on reviewer’s suggestion, we omitted the word “in” from the sentence. Please refer line 35 for the change.

COMMENT# 2:

line t2: ...is the group of heat shock proteins… (word "One is single", "proteins is plural)

Response to Comment #2:

As per reviewer’s remark, we corrected the sentence. Please refer line 52 for the change.

Afterwards, we appreciate your valuable suggestions. Please let me know if we did not answer all of your comments clearly or if you require any further information.

Best regards

Kwanseob Shim, PhD

Round 3

Reviewer 3 Report

The authors have tried their best to revise the current manuscript. The primer design need to be improved in the future related research. The current primer is still overlap with some of other primers (ethanolamine kinase 1).

Even though, some of parts still need improvement but the manuscript is good enough to provide the audience with useful information. I recommend: Accept

Author Response

Kwanseob Shim, PhD

                                                                   Dept of Animal Biotechnology

Jeonbuk National University

Jeonju, 54896.

Republic of Korea

Vladislava Sesum,
Animals Editorial Office
MDPI Novi Sad Office
Arse Teodorovica 15, 21000 Novi Sad, Serbia

Dear Vladislava Sesum,

The enclosed manuscript is our revised manuscript entitled “Acute heat stress induces differential expression of heat shock proteins in different sections of the chicken small intestine based on exposure duration” by S H Siddiqui et al for publication in Animals (Manuscript number: animals-778500).We appreciate the time and efforts by the editor and reviewer of Animals in reviewing our manuscript. We corrected and answered all of points addressed by the reviews, and believed that the revised version can meet the publication requirements of Animals. We are deeply thankful for your helpful comments and suggestions. These comments have helped us a lot in clarifying and sharpening our manuscript. Following your suggestion, we have modified the manuscript accordingly, and detailed corrections are listed as below.

Point-by-point response to Reviewer #3

COMMENT# 1: 

The authors have tried their best to revise the current manuscript. The primer design need to be improved in the future related research. The current primer is still overlap with some of other primers (ethanolamine kinase 1).

Response to Comment #1:

First of all, we would like to express our gratitude to the reviewer for reviewing our manuscript thoroughly. We will be more sincere and careful to design our primer in the future related research. The current primer has been checked by NCBI Primer-BLAST. We attach a screenshot of the primer sequence (Please see the attachment).

We have improved most parts of our manuscript according to all reviewers’ comments, so we hope it now matches the journal standard. Afterward, we would like to convey our thanks for acknowledging our manuscript as providing useful information to the audience as well as recommending it as “accept” for the journal.

Finally, we appreciate your valuable suggestions. Please let me know if we did not answer all of your comments clearly or if you require any further information.

Best regards

Kwanseob Shim, PhD
